# Determining Indicators Related to Land Management Interventions to Measure Spatial Inequalities in an Urban (Re)Development Process

**Youness Achmani [1],*, Walter T. de Vries [2] , José Serrano [1] and Mathieu Bonnefond [3]**

1   Département Aménagement et Environnement, Ecole polytechnique de l'Université de Tours,
    35 allée F. de Lesseps, 37205 Tours, France; jose.serrano@univ-tours.fr
2   Lehrstuhl für Bodenordnung und Landentwicklung, Ingenieurfakultät Bau Geo Umwelt,
    Technische Universität München, Arcisstrasse 21, 80333 München, Germany; wt.de-vries@tum.de
3   EA 4630 Laboratoire Géomatique et Foncier, Conservatoire National des Arts et Métiers, HESAM Université,
    1 Boulevard Pythagore, 72000 Le Mans, France; mathieu.bonnefond@lecnam.net
*   Correspondence: youness.achmani@etu.univ-tours.fr; Tel.: +33-753-01-56-82

**Abstract:** Nowadays, urban sprawl, urban densification, housing shortages, and land scarcity are some problems that intervene in the practice of urban planning. Those specific problems are currently more than ever emergent because they imply the notion of spatial justice and socio-spatial inequalities. Hence, it seems necessary to promptly research and describe these from a new and different perspective. Thus, we consider the Institutional Analysis and Development to define a conceptual framework to assess spatial justice. We simplify it into a three-dimensional model (rule, process, and outcomes) in which a matrix of indicators applies on each level. We elaborate the indicators to measure spatial inequalities in an urban development project, for which the reason we refer to the egalitarian paradigm of spatial justice. While spatial inequalities raise questions about land management, we elaborate those indicators related to three land management interventions: the use, access to, and redistribution of land use.

**Keywords:** spatial justice; spatial inequalities; Institutional Analysis and Development; land management; urban (re)development process

## 1. Introduction

The increase in urban dynamics revives the debate on the Just City, the right to the city and spatial justice. Among these, we find the housing shortage and the problematic of providing social and affordable housing [1–3], urban expansion [4–7] and land scarcity [8]. Those problems of urban planning are currently more than ever emergent and can be reviewed from different perspectives but we aim to take a more conceptual approach, because this is currently missing in the contemporary debate. For this purpose, we refer to the egalitarian paradigm of spatial justice to develop indicators that measure spatial inequalities, i.e., appropriate indicators that arise from the principle of equity discussed in the Just City [9,10], the right to the city [11] and the Theory of Justice [12]. Hereby, we consider the egalitarian paradigm of spatial justice and exclude the libertarian one where the notion of inequalities does not represent an issue.

Measuring spatial inequalities raises questions about land management, because how the urban planner acts on land raises questions about what is the right thing to do and hence could be discussed from the lens of spatial justice. For this reason, we aim to develop those indicators in relationship with three land management interventions: the use, access to and redistribution of land use. By use, we refer to the way one can develop the land, such as public amenities and facilities, housing and offices or

green space. Access to land refers to the ability to control the space resources covered by the land and to transfer the rights to the land. Finally, the redistribution is related to the manner in which a planner makes rational decisions about the allocations and divisions of space and equitable access to land is promoted [13].

Thus, in order to construct a conceptual framework to assess spatial justice, we refer to the Institutional Analysis and Development (IAD) [14,15] because it implies the question of spatial justice from its component "evaluative criteria" and also because it is suitable for the question of land management [16]. We simplify the IAD framework into three components: rules, processes and outcomes. Rules represent the inputs or the exogenous parameters, which include biophysical conditions, attributes of community, and rules in use. This is to refer to what Barrie Needham [17] defines as rules of the game and rules on all. The first includes the rules in use and arise from structuring the set of rules of game, whether they choose to play or not. The second emerges from regulation, which influences how people may operate within the market rules, and can restrict people in the exercise of their rights in an additional way, a way—moreover—which people cannot avoid by deciding not to take part in a transaction, and includes discussing two types of rules: regulation and structuring. Then, we attribute processes to what takes place in the action arena, namely the actions and interactions between participants, the steps, conditionalities and sequences to make decisions within a situation affected by the physical, community, and institutional characteristics that will then result in varying patterns of interactions and outcomes [18–22]. The outcomes are often the result of numerable actions. The Evaluative Criteria from the IAD assess the likely set of outcomes that could be achieved under alternative actions or institutional arrangements [23,24].

This paper starts by reviewing the paradigms related to spatial justice. This is followed by an evaluation of how the Institutional Analysis and Development framework could be applied to design an indicator framework. This derives a basic framework of spatial justice dimensions. This framework is then applied to a generic context of urban development projects in order to determine a set of indicators related to the three dimensions of land management interventions. Considering that studies from real life problems help to build the matrix of indicators and allow to assess the validity of the framework and possibly alter, detail or improve it. We conclude with a summary of the findings and an outlook for further research.

## 2. Theoretical Background

Literature on spatial justice has two main entry points: on the one hand, it can be approached from the perspective of how and why to provide an equitable distribution of resources (land), in such a way that it tackles spatial inequalities. This is the egalitarian vision of justice, in which the principle of equity is fundamental. This implies taking action against socio-spatial inequalities by promoting better territorial organisation and a more equitable distribution of resources (public amenities, affordable housing, services public, sports and cultural facilities) in the available space [25]. On the other hand, libertarians define spatial justice as the distribution of wealth by trickle-down: this understanding of spatial justice is the pseudo-scientific justification of "neoliberal theories 'promoting a distribution of wealth by' runoff" [26]. Here, the wealth underlies the allocation of resources following the market (demand and supply) and laissez-faire from the state [27] instead of an interventionism to promote the access and the allocation of land for poor and low-income people.

### 2.1. Libertarians Do Not Consider Spatial Inequalities an Issue

Western political philosophy from Plato to Hegel via Aristotle, Machiavelli, Hobbes, Spinoza, Montesquieu, etc., has repeatedly justified the existence of inequalities of fortune, power or culture. They legitimate common social inequalities between people. In the same vein, the majority of current non-egalitarian discourses, related to the current ideas of neoliberalism, does not consider social inequalities as a problematic issue. For instance, libertarians do not give a special place to the expectations of the most disadvantaged [28]. In fact, for any inequalities in a social situation (of income,

wealth, power, etc.), neoliberalists would consider these either as essential, negligible, or as the price to be paid for the guarantee of economic efficiency and political freedom [29].

For the authors of the neo-Marxist movement, the process of neoliberalisation is associated with a retreat of redistributive and procedural justice, or even with a form of authoritarianism. It is geographically unequal and socially regressive [30] which is "sometimes" regarded as a form of injustice [31]. Hence, libertarian thinking has no consideration for social inequalities and eventually spatial inequalities because neglecting the first leads to neglecting the second[1]. For instance, a space could be equal (by responding to indicators constructed below) though it does not address social inequalities (middle and poor class will remain the same).

### 2.2. Considering the Egalitarian Paradigm to Assess Socio-Spatial Inequalities

The egalitarian paradigm is concerned by the principle of equity, which could for example be found in the theory of the Just City [10] the right to the city [11] and obviously in the contractarian theory of justice [12]. In her theory, Suzan Fainstein [10,32] defines a set of expectations that ought to form the basis for just urban planning. She enumerates equality, diversity and democracy, and their contents apply only to planning conducted at a local level. In the furtherance of equality: all new housing developments should provide units for households with incomes below the median, with the goal of providing a decent home and suitable living environment for everyone. Planners should take an active role in deliberative settings in pressing for egalitarian solutions and blocking ones that disproportionately benefit the already well-off. In furtherance of diversity, zoning should not be used to further discriminatory ends and uses should be mixed. At the same time, ample public space should be widely accessible and varied and be designed so that groups with clashing lifestyles do not have to occupy the same location. In the furtherance of democracy, plans should be developed in consultation with the target population if the area is already developed. The existing population, however, should not be the sole arbiter of the future of an area.

Those principles are corroborated in Rawls' theory of justice, where he uses the term equity. In fact, he sets out the principles of justice regarding a fair distribution of primary social goods (rights, freedoms, opportunities offered to individuals, wealth, income). In Rawls' liberal–egalitarian conception (where the two fundamental principles are freedom and equality), justice (or "justice as equity") must satisfy two principles [12]. The first one concerns fundamental freedoms, whilst the second is related to opportunities in access to various social positions and socio-economic benefits. The latter states that social and economic positions are to be (a) to everyone's advantage, especially for those who have less (maximin) and (b) open to all in accordance with the fair equality of opportunity. In fact, urban projects influence the opportunities of access in another way. Among these opportunities, one thinks more spontaneously of income or education conditions, but the ability (or inability) to move around, to find housing, to access efficient health care, and to benefit from a comfortable living environment, also contribute to reinforcing the chances of access to various social positions. Through its implementation, an urban project probably has little effect on the first possibilities (income, education), but it has an undeniable role on the following ones: possibilities of movement, housing, access to health care. Indeed, this is what Henri Lefebvre [11] calls the right to the city, in which he encompasses the right to work, to training and education, to health, housing, democratic participation in decision making, etc.—the necessities for a decent life [33].

Thus, Lefebvre's concern was not to propose a new comprehensive slogan demanding the right to basic needs. It was about something more—a specific urban quality, which had hitherto been neglected in public debate: access to the resources of the city for all segments of the population, and the possibility of experimenting with and realising alternative ways of life [34]. The right to the city is a moral

---

[1]  In fact, the relationship between social and spatial inequalities could be assessed in the definition of socio-spatial disparities [27]: the differences between geographic areas regarding the socio-economic characteristics of their inhabitants.

claim, founded on fundamental principles of justice. "Right" is not meant as a legal claim enforceable through a judicial process today, rather, it is multiple rights that are incorporated here: not just one, not just a right to public space, or a right to information and transparency in government, or a right to access to the centre, or a right to this service, but the right to a totality, a complexity, in which each of the parts is part of a single whole, to which the right is demanded [34]. Harvey [35] formulates well what such a city/society might be in principle; he uses Robert Park's phrase "the city of heart's desire": the principles that such a city would incorporate can be set forth in general terms. They would include concepts such as justice, equity, democracy, beauty, accessibility, community, public space, environmental quality, support for the full development of human potentials or capabilities, to all according to their needs, from all according to their abilities, and the recognition of human differences. They would include terms such as sustainability and diversity, but these are rather constraints on the pursuit of goals rather than goals in themselves.

While the definition of social inequalities is based on the mathematical notion[2] of inequality, it is no less inevitable that it should refer to the feeling of injustice that they create, among those who obviously suffer from but also possibly among other members of society. By analogy, looking at socio-spatial inequalities means considering them as intolerable and unjust: in other words, as a vector of spatial injustice. Thus, concentrating investments in a few carefully selected areas contributed to the intensification of socio-spatial inequalities and the fragmentation of cities [26,36,37]. Suzan Fainstein [10] states that it is way too easy to follow the lead of developers and politicians who make economic competitiveness the highest priority and give little or no consideration to questions of justice. Thus, the egalitarian paradigm seems consistent to measure spatial inequalities, but it remains a set of ideas and challenging to implement in an urban project. In the following sections, we suggest exploring it to elaborate indicators to each three-dimensional framework level.

*2.3. Evaluation of Spatial Justice Based on Egalitarian Paradigm*

In order to measure spatial inequalities, we are considering the egalitarian paradigm of justice. Based on the three-dimensional model of spatial justice, and the evaluative criteria of equity from the IAD framework, we could define the rule, process and outcomes as per Table 1.

**Table 1.** Defining the three dimensions of spatial justice.

|  | **Definition** | **Bibliographic References** |
|---|---|---|
| Rule | Rules provide all people with equal opportunities to access and/or use spatial resources. | [21,38,39] |
| Process | Consists of designing and implementing plans and activities that pertain to the management of space through active participation and collaboration among users of spatial resources, decision makers and planners. | [10,40–46] |
| Outcomes | The access to spatial resources, their uses, and their redistribution on the space. | [40,44,47] |

Addressing spatial justice and seeking spatial justice presupposes the institutionalisation of rules which provide all people with equal opportunities to access and or use spatial resources [38,48]. Rawls [39] concludes that individuals would choose a system of equal opportunity, which involves "a framework of political and legal institutions that adjust the long-run trend of economic forces so as to prevent excessive concentrations of property and wealth, especially those likely to lead to political domination".

---

2　　In the sense of non-equality, mainly in term of incomes.

The process and outcomes describe, respectively, the procedural and structural (redistributive) justice. Susan Fainstein [10] explain that "among planning theories there is a debate between those who emphasise communication, negotiation, and democratic decision making as the principal normative standard for planning and those who instead opt for a substantive concept of justice". In the same vein, Patsy Healey [49] highlights that procedural and structural justice are not two separate spheres, but rather are complementary. Frank Fisher concomitantly states that "the debate is unproductive and that the two points of view can be brought together within a broader framework" [50]. He continues "for the communicative theorists, the test of policy depends on who is included in its formulation, on the existence of an open, fair process, and on better argument as the deciding factor. For just-city theorists, the principal test is whether the outcome of the process (not just of deliberation but of actual implementation) is equitable; values of democratic inclusion also matter, but not as much" [50].

In fact, the processes consist of designing and implementing plans and activities that pertain to the management of space through active participation and collaboration among users of spatial resources, decision-makers, and planners [44]. Iris-Marion Young [45], Chantal Mouffe [46], Mark Purcell [42] and Suzan Fainstein [10] emphasize a democratic and more open process, and lay down the supporting arguments for communicative model developed within planning theory (communicative planning). They maintain that a pluralistic, decentred form of democratic participation, in which social movements press strong demands, will incorporate equity goals and thereby reduce injustice. According to Iris Marion Young [40], Colin Crouch [41], Erik Swyngedouw [43] and Susan Fainstein [10], fair urban policies cannot be produced if the dominated social groups are not involved in their production process. In other words, inclusive urban governance is necessary for the production of just urban policies. Finally, the outcomes are evaluated through the analysis of the results of the dialectical processes of the production and reorganisation of the space. The outcomes cover the aspects of peoples' relations to space. They include the access to spatial resources, their ownership, their uses, and the inhabitancy of the space [44,47].

## 3. Research Methodology

Using the definitions of Table 1, we will develop indicators related to land management interventions in each level (rule, process and outcomes). The development of evaluative indicators follows a deductive research approach and based at different scales: the macro, meso, and micro levels but also in cities in different countries. Those levels are absolutely observed for the development of indicators that evaluate any development program [51,52]. The macro level relates to a high level, such as the regional or national scale. The meso level refers to the municipality, district, or another low level while the micro level relates to the household or individual scale. Those levels can be aggregated at other hierarchies depending on the geographical scale of the study [53,54]. For this reason, the indicators will be aggregated at the scales of the city (macro), urban neighbourhood (meso) and household (micro). It means that any of the proposed indicators can measure spatial justice at one or more levels among the urban project.

In order to define an institutional framework to assess spatial justice, we used the Institutional Analysis and Development framework. The IAD is considered as a framework and not a theory [19–21] and encapsulates the collective efforts of this intellectual community to understand the ways in which institutions operate and change over time. The IAD framework assigns all relevant explanatory factors and variables to categories and locates these categories within a foundational structure of logical relationships [22]. Figure 1 delineates the different components of the framework.

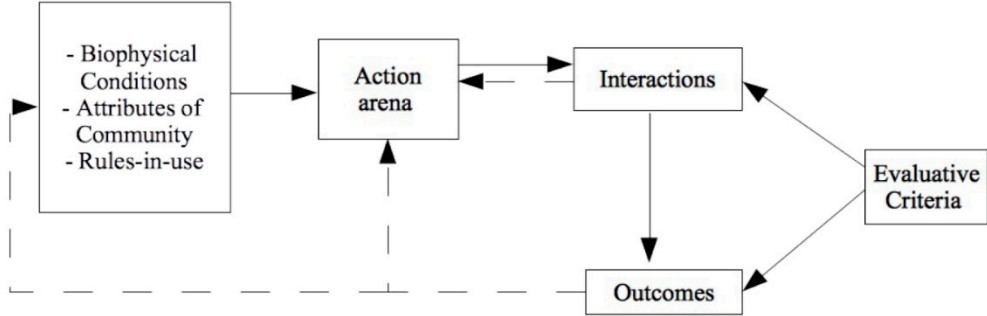

**Figure 1.** Institutional Analysis and Development framework. Developed based on [16,19–22,24].

The framework is suitable to question the spatial justice. In fact, the evaluative criteria take into consideration the notion of justice via the criteria of equity and efficiency. Our approach involves the egalitarian paradigm and thus makes use of equity and social justice principles. This is in line with what Suzan Fainstein [9,10,32] enhances: "there is not always a trade-off between justice and efficiency, but when there is, the demands of justice should prevail". In the following sections, we will build our model of spatial justice by dividing it into three levels: rule, process and outcomes. Then, and by implementing the evaluative criteria, we will define the three-dimensional framework of spatial justice in which a set of indicators apply on each level.

### 3.1. Approaching the IAD Framework by Three Dimensions: Rule, Process and Outcomes

Three levels could be perceived: the input, action arena and outcomes. The input contains the biophysical conditions, the attributes of community and the rules in use. Those parameters refer to the factors influencing on the action arena [16], we considered them as the rule level. The action arena focuses on how people cooperate or do not cooperate with each other in various circumstances. The analysis needs to identify the specific participants and the roles they play within the situation. It will look at what actions have been taken, can be taken or will be taken and how they affect outcomes [18]. Indeed, it refers to the process in which actors participate to define outcomes: "outcomes are generated by the conjuncture of the outputs of a given action situation, other closely related action situations, and exogenous influences that may not always be subject to effective control of human intervention" [22]. Figure 2 shows the relationship between the three-dimensional model and the IAD framework.

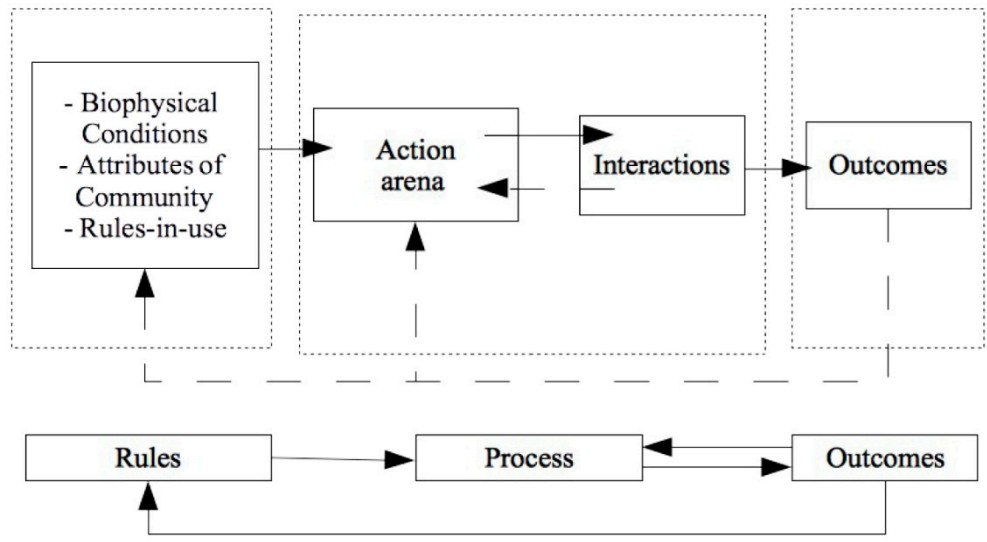

**Figure 2.** Approaching the IAD framework by three-dimensional model rule-process-outcomes.

### 3.2. Defining an Institutional Framework for Spatial Justice

Another component apply on outcomes [18,23] is "evaluative criteria", which contain some of the most frequently used criteria: (1) increasing scientific knowledge, (2) sustainability and preservation, (3) participation standards; (4) economic efficiency, (5) equity through fiscal equivalence, and (6) redistributional equity [18]. Related to justice, both equity and efficiency are relevant to consider and may be used "be used by participants or external observers to determine which aspects of the observed outcomes are deemed satisfactory and which aspects are in need of improvement" [22]. However, even if they allow us to evaluate outcomes that are being achieved as well as the likely set of outcomes that could be achieved under alternative actions or institutional arrangements [15,22], they are also involved to evaluate the process and the rules.

While efficiency would dictate that scarce resources be used where they produce the greatest net benefit, equity goals may temper this objective, resulting in the provision of facilities that benefit particularly needy groups. Redistributional objectives tend to conflict with the goal of achieving fiscal equivalence[3], and tough decisions as to which aspect of equity needs priority must be made [18]. Indeed, the two evaluative criteria refer to the paradigms of spatial justice: on the one hand, the principle of efficiency formulates the view of libertarians while equity stands for the egalitarians. Thus, we obtain the three-dimensional model of spatial justice [10,15,38,40,47,55–58] in which indicators are applied on the rule, process and outcomes. Our framework could be presented as per Figure 3.

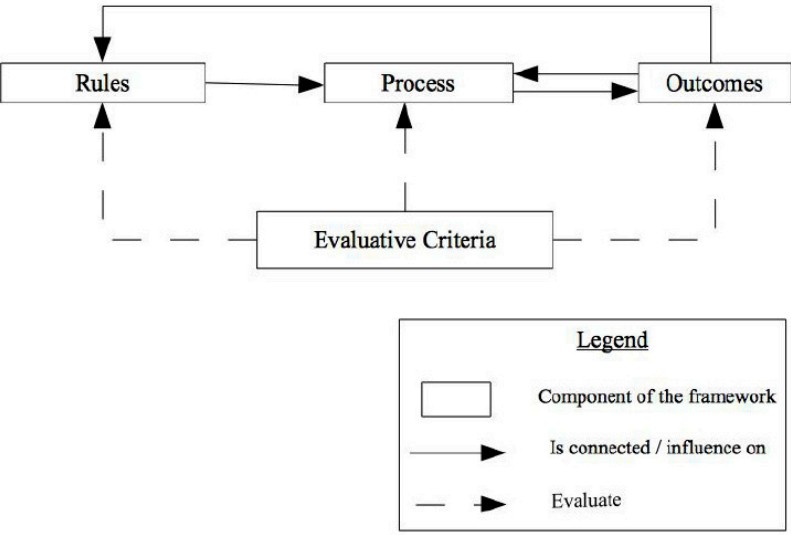

**Figure 3.** Institutional Analysis Framework to assess spatial justice.

The relationship between the three dimensions is reflected by the arrows, which stipulate that rules provide the guidelines for the processes from which the outcomes derive. According to the theoretical framework of spatial justice, there are arrows that link outcomes to rules and outcomes to process, which asserts that we can always take a step back and operate on process or rules in order to showcase spatial justice.

### 4. Results and Discussion

Taking into account the three components of Table 1, it is now necessary to test how this could be applied in a real-life context. We are interested to measure spatial inequalities in an urban (re)development project. By this notion, we refer to different processes of urban (re)development that aim to develop a piece of city by regeneration, intensification or creation ex nihilo [59]. This include the

---

[3] "The extent to which the beneficiaries of a public good or service are expected to contribute towards its production" [22].

most important reviewed processes like housing development, sites and services schemes, and urban regeneration in its different aspects[4]. Sites and services schemes stand for the process of subdividing public land or land acquired through public funds into buildable lots. Those lots are improved through the provision of basic infrastructures and services. Thereafter, they are allocated to poor and low-income people at low prices for self-housing development. Urban regeneration refers to the process of transformation, renewal of a part of the city (wasteland renewal, relocation of industry from the inner city, etc.).

### 4.1. Determining Indicators Related to the Use of Land

Typical urban development projects contain the following elements and characteristics (Table 2). The principle of social mix is privileged, i.e., the project has to merge between affordable and free-financed housing. Indeed, the rate of affordable housing indicates how the project is dealing with social segregation [10,60,61] but not sufficient to question socio-spatial inequalities [32,62–65]. For instance, a program which aims at a particular social class is a vector of socio-spatial injustice, even if it only produces affordable housing. However, assessing spatial justice is beyond social mix or improving housing policy but also a functional mix which means that the project has to offer public amenities, public services, green spaces, etc. In this context, there is a need to consider the project in different scales, starting from the program and its neighbourhood, to the city and the metropole. Table 2 summarises the connection between indicators and the three level of spatial justice.

**Table 2.** Indicators related to use of land.

| Land Management Intervention | Indicator | Connection to Spatial Justice | | | Indicative References |
|---|---|---|---|---|---|
| | | Rule | Process | Outcomes | |
| Use | Functional mix | Strategies and rules for improving functional mix;A zoning including different amenities | Insert the project in large scale and consider the needs in the neighbours; integration of revitalized areas and their inhabitants into the modern city | Promotion of good-quality facilities, services, and housing | [32,64–67] |
| | Existence of public equipment and public services | | | Provision of basic services and infrastructure: public equipment and public services | |
| | Social mix[5] | Rules and strategies to improve and emphasise social mix | Inclusion into a zoning scheme of affordable units for poor and low-income groups | Promotion of access to houses for all people and elimination of inequalities in housing; integration of poor and low-income groups into the city | [10,62,63,68] |

Social mix is defined as mixed-income housing that either requires or encourages income-integrated residential developments through the inclusion of low-income housing units as part of a market-rate project [66–70]. The notion could be expressed in different ways: mix tenure [71], inclusionary zoning [68] or mixed communities [65]. The social mix is usually related to social segregation that was debated in different studies [60,61,65,66,71] and showed its limits regarding the question of spatial justice. In fact, Michelle Norris [71] states that tenure mixing in urban areas is of particular importance to avoid undue social segregation. In the same vein, the "inclusionary zoning programs may serve to

---

[4]　It includes housing renewal, urban transformation, wastelands renewal, etc. Housing renewal consists of improving the physical, social, economic, and ecological aspects of old urban neighbourhoods. The process involves also the revitalisation of individual or community properties, including dwelling units. This gives the local community options to renovate their existing buildings or demolish them in order to develop new ones.

[5]　Or mix tenure, or inclusionary zoning, or mixed communities (McIntyre).

limit residential racial segregation and discrimination by creating housing opportunities for a range of incomes and by counteracting the exclusionary effects of restrictive land use regulations [66,72–75]. In the face of growing urban segregation and the idea that working-class neighbourhoods have become "places of relegation ", the aim of social mixing is to mix socially diverse populations in a given area ( . . . ) and to re-establish "republican integration" [60]. The social mix is also defended by Suzan Fainstein [10] in order to gain more equity: "uses should be mixed".

The concept of functional mix has been discussed by Van de Walle et al. [64] and refers to two frames of reference. On the one hand, the sustainable development reference aims to reduce travel and promote the "city of short distances," dense, mixed, and economical in space and energy. In this reference, we found the ideas of McIntyre and McKee [65] that highlight "the discourse of sustainable development has become one of the central orthodoxies of planning. This notion is based on the idea that some places have become 'unsustainable' ( . . . ) the Social Exclusion Unit quickly identified areas with high levels of deprivation". On the other hand, the egalitarian reference intends to reduce spatial inequalities and improve the inhabitants' lives. It occurs by inserting into underprivileged and mono-functional neighborhoods, mainly large housing estates, facilities, services, economic activities, shops, etc. Both references are against spatial specialization (housing, work, travel, entertainment), which define the functional mix [63] and aim for spatial equality.

## 4.2. Determining Indicators Related to Access to Land

Land markets are (virtual and physical) spaces where individuals and firms compete for access to land. Thus, like with usual markets, demand and supply also drive them. On the demand side, there is population growth, while on the supply side, there is the annual production of serviced land, and the amount of land that is not available in the market and zoning that informs the land use for a specific parcel of land [76]. Considering housing as an important asset in the land market, as the prices are primarily dependent on the allocated land use and the location from the city centre [77,78], and thus land markets play an important role in the distribution of different socioeconomic groups [79]. In addition, the value of land is increased for the benefit of landlords due to the existing suprastructure, mostly when the project is located in the inner city (public transport infrastructure, schools, etc,). It is evident that both society and the government play a relevant role in influencing the value of land [80,81], and by considering the principles of equity, "those who benefit from a service should bear the burden of financing that service" [18], which means a need for a land value capture. In this way, the state could develop affordable housing that is sold or rented at low prices. This helps poor and low-income groups mitigate the problems of non-access to shelter [45]. In the same vein, Fainstein [10] recommends "all new housing developments should provide units for households with incomes below the median, with the goal of providing a decent home and suitable living environment for everyone". This is what Peter Marcuse [33] emphasises from his interpretation of the right to the city [11], "the concerned right can be itemized: the right to clean water, clean air, housing, decent sanitation, mobility, education, health care, democratic participation in decision making, etc".

At the same time, access to land does not only relate to a housing market but also to public amenities and public services. In this context, the notion of scale is important for a spatial planner, who is required to think of the project beyond the locational necessities of the neighbours, the city or even the metropole. The access to land could be found in the Christian Schmidt interpretation [34] of the right to the city: "access to the resources of the city for all segments of the population". Moreover, public transport plays a crucial role to promote access to land. In an urban area, mostly the supra-infrastructure is already existing but sometimes it needs to be improved. Hence, Table 3 illustrates the indicators related to access to land.

**Table 3.** Indicators related to access to land.

| Land Management Intervention | Indicator | Connection to Spatial Justice | | | Indicative References |
| | | Rule | Process | Outcomes | |
|---|---|---|---|---|---|
| Access | Access to public amenities and public services | Rules promoting the access to the city | Participation of and collaboration with all categories of people in spatial planning; they permit the integration of their needs and rights into urban development programs | Promotion of access to land resources or other urban amenities for all people, including poor, vulnerable, and low-income groups | [10,11,33–35] |
| | Housing | Inclusion into a zoning scheme of affordable units for poor and low-income groups | Integration of poor and low-income groups into the urban fabric | Promotion of access to houses for all people and elimination of inequalities in housing | [10,11,33,34] |

*4.3. Determining Indicators Related to Redistribution of Land-Use*

Measuring socio-spatial disparities questions the spatial distribution of public amenities and housing in the space. At a different scale, the project needs to avoid the monostructing of housing, and to balance between the distribution of affordable housing and the location of equipment. At a larger scale, the project has to take into consideration the needs of public amenities among the neighbours, the city and the metropole. "The [resolution of] concentrated poverty 'problem' has considerable implications for socio-spatial justice and the geopolitics of life chances. One particular governmental solution (...) in the housing arena has been the creation of mixed communities" [65]. This is also a way to face social segregation: "the complete separation of social housing ( . . . ) might undermine the aim of mitigating social segregation" [71]. In the same vein, Kontokosta [66] calls for an inclusionary zoning that respects an equitable distribution, and not only social mix: "for inclusionary zoning to be an effective policy tool, it must not only create new affordable housing, but also spatially disperse new units and promote income and racial integration within a specific community". In fact, "one of the original goals of inclusionary zoning is income integration at the project level and, by extension, the surrounding neighbourhood, thus avoiding the low-income concentrations historically created by other subsidised housing programs" [68,69]. This is what Suzan Fainstein [10] also emphasises "ample public space should be widely accessible and varied but be designed so that groups with clashing lifestyles do not have to occupy the same location".

The urban projects in the inner city are profitable and mainly a vector of displacement, so the highest and the best use of land will leave the powerless and vulnerable groups in dire state of poverty and exclusion alongside diminishing social goods. In this context, the government has an active role to ensure an equitable distribution and at the same time to enforce a social mix [82]. Suzan Fainstein [10] states "what is important is that people are not differentiated and excluded according to ascriptive characteristics like gender or ethnicity. But neither should people be required to tolerate disorderly conduct or anti-social behaviour in the name of social justice".

At the same time, the project has to carry the location of public amenities in a way to serve a large number of users. This highlights the integration of the project at a large scale, what Suzan Faintein [10] defends as: "citywide considerations must also apply". This is also what Harvey states [35] when he uses Robert Park's phrase "the city of heart's desire". Hence, Table 4 includes the different indicators related to land redistribution, in which it includes the definition of each level from the spatial justice framework.

**Table 4.** Indicators related to the redistribution of land.

| Land Management Intervention | Indicator | Connection to Spatial Justice | | | Indicative References |
| | | Rule | Process | Outcomes | |
| --- | --- | --- | --- | --- | --- |
| Redistribution | Distribution of public amenities | Strategies and rules improving access to resources for a large number of users (residents and others) | Participation of and collaboration with all categories of people in spatial planning and adapt the redistribution of land use to their needs | Equitable redistribution of public amenities; promotion of access to good-quality facilities, services, and housing | [10,35] |
| | Distribution of affordable housing | Rules and strategies to avoid segregation; inclusion into a zoning scheme of affordable units for poor and low-income groups | | Equitable distribution of affordable housing; elimination of inequalities in housing | [67,68,82] |

The institutional framework developed for spatial justice takes into consideration the egalitarian paradigm which guides us to elaborate indicators. Additionally, it helped to define the several dimensions of the model, rules, process and outcomes. Indeed, the analytical framework in question is already developed and used in several studies [10,11,15,35,38,40,56,57], particularly in the study of Uwayezu and de Vries [58], in which they used the model to measure spatial justice and land tenure security. However, the indicators developed in this paper are related to land management interventions because spatial justice is also land management problematic. Moreover, the definition of process and outcomes refers, respectively, to procedural justice and structural one. This is different from the previous study of Uwayezu and de Vries [58] in which they establish indicators from urban (re)development rules, processes, and outcomes on an ordinal scale with four levels. Those levels range from very high (for indicators that relate to procedural justice), high (for indicators that relate to recognition justice), and moderate (for indicators that relate to redistributive justice) to low (for indicators that relate to intra- and inter-generation justice). It means that we can find an indicator that evaluates the process and, at the same time, is related to redistributive or procedural justice.

This framework concerns an urban (re)development project, and is applicable only to urban studies but not rural ones. However, the basic three-dimensional model to assess spatial justice could be applied in every urban (re)development project. For further studies, we can improve it to be implemented in rural areas, thus the indicators need to be related to land justice, what we can find in a some studies like Baysse-Lainé [82] or recommendations from the International Land Coalition [83]. Finally, this paper will be followed up by a more quantitative approach to measure each of the indicators in multiple case studies. Moreover, we will proceed by a qualitative study so as to compare if certain issues are generic or idiosyncratic.

## 5. Summary and Conclusions

This paper proposes a conceptual approach to study the spatial justice and measure spatial inequalities. In fact, there is a relationship between spatial justice and spatial inequalities that could be assessed by the egalitarian paradigm. The latter defines spatial justice as the equitable redistribution of wealth (land) and the addressing of spatial inequalities. The ideas behind the egalitarian paradigm are discussed in Theory of Justice [12], the Just City [10] and the right to the city [11] but cannot lead to an analytical framework to assess spatial justice. For this reason, we used the IAD framework to develop an institutional framework, what we simplified it into a three-dimensional model: rule, process and outcomes. This model is already developed in previous studies in which a matrix of indicators applies at the different levels. While spatial inequalities raise questions about land management, we defined the matrix of indicators related to three dimensions that affect land management interventions: the use, the access to and the redistribution of land use.

The three-dimensional framework discusses procedural and structural justice in one model, in which the process and outcomes refer to them, respectively. This paper considers them as two

integrated spheres in which the rules take part as the first level to address spatial justice. They are essential because they give actors legitimacy to argue and participate during the process [84]. They are connected with the outcomes because if an actor is not satisfied with a non-ultimate result, he could continually use the rules to influence the results.

This paper also discusses also the urban (re)development processes that have potential to promote/prevent spatial justice and socio-spatial inequalities. This includes the development of affordable or mixed housing, urban regeneration, or in general the urban (re)development that aims to produce a piece of the city. To assess whether these approaches and processes promote spatial justice and spatial equality, this paper proposes a holistic set of seven evaluative indicators. Among them, three are connected to the use of land, two to the access and allocation of the space to people and two to the redistribution of land use. For each one, we define it in relation to the dimension of spatial justice, i.e., the rule, process and outcomes.

All proposed indicators are defined according to the egalitarian paradigm. The evaluation of spatial inequalities using those indicators can be performed at different stages of an urban project. The users of those indicators can include decision makers, municipality authorities, urban planners, and different organisations who intend to measure trends of spatial justice and spatial inequalities in the course of any urban (re)development programme. However, this paper does not provide a quantitative approach or an evaluation test of the framework using a specific case study. Instead, it shows that it is possible to create an analytical framework of indicators that addresses multiple themes. The extent to which, the conditions under which, and the kinds of cases for which the framework can be used practically, will be the aim of a subsequent publication.

**Author Contributions:** This manuscript is a part of Y.A.'s ongoing Ph.D. research. All authors set up the structure and approach of the manuscript. Y.A. contributed to all sections under the guidance of W.T.d.V., J.S. and M.B. They all made a large contribution to the revision and editing of the whole manuscript. W.T.d.V. supervised the work during the stay of Y.A. at TUM, while J.S. and M.B. are the supervisors at Y.A.'s home university. All authors have read and agreed to the published version of the manuscript.

**Funding:** This research received no external funding.

**Conflicts of Interest:** The authors declare no conflict of interest.

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
