# Peer review of "Determining Indicators Related to Land Management Interventions to Measure Spatial Inequalities in an Urban (Re)Development Process"

_land, doi:10.3390/land9110448_

Round 1

Reviewer 1 Report

The considerations presented in the article concern the conceptualization of spatial inequalities in an urban project. The authors' proposal is universal, i.e. it can be applied not only at the micro-, meso- and macro-level, but also in cities of different countries. Hence, I believe that the discussed topic is interesting and worth developing.

I see two strengths of the article. The first is an attempt to create an original model of spatial inequalities measurement. I accept this goal with satisfaction, especially since there are not many studies on this subject in the literature. The second is a very rich literature review. A total of 91 studies were used, which is impressive for one article.

There are also weaknesses that should be the basis for reflection for the authors. The first weakness is the wrong structure of the article. Chapter 2. Research methodology, should be placed after 3. Theoretical background. Moreover, the last paragraph of chapter three (p. 7) "Using the definitions of table 1 ..." is the setting of the research goal, and therefore should be included in the chapter “Research methodology”. The other weakness is the lack of sufficient critical reflection on literature and their own conception. As a reader, I had the impression that authors chose literature to prove their concept. It is not about the number of literature, because it is high and constitutes the strong point of the article. It is about the authors' reflection on whether the cited literature has limitations that could adversely affect the effectiveness of one's own concept. One of the necessary directions for improving the article must be a critical approach to the possibility of implementing the "social mix". The authors' proposal does not convince me that it can be effective in achieving a "social mix". I suggest that the authors of the article answer the questions how to achieve a "social mix"? Will the middle-class and the rich relocate to neighborhoods where poor residents are (or will be) located? After all, we know from Urban Economics that social pathologies are correlated with households with incomes below the median. The authors of the article only highlight this problem on p. 10: "But neither should people be required to tolerate disorderly conduct or anti-social behavior in the name of social justice"; but they do not develop this problem. Additionally, it should be noted that in order for a "social mix" to be effective, it must be in the right scale. If there are not enough housing available for low incomes households, then the concept will be ineffective. What is the rate of such households? It is true that the authors conclude that future research will be quantitative, but the basic statistics from selected cities could verify the authors' model. Here I mean the numbers of the city's population structure by income.

All in all, the article is interesting. Moreover, it can be a frequently read and quoted study because it deals with a topic that is universal for a large number of cities in the world. Before the article is published, however, the authors should consider the above-mentioned questions. The point is to convince the reader that the concept of the authors result from a critical analysis of the literature, and not from proving a preconceived thesis.

Reviewer 2 Report

Summary

The article introduces a conceptual framework to assess spatial justice based on a three-dimensional model (rule, process and outcomes). Indicators in order to measure spatial inequalities related to three management interventions (use, access to and redistribution of land-use) are presented.

Overall comments:

The proposed simplified model is a good starting point and approach to access spatial inequalities in an urban project. The presented indicators are proper. Literature applied is sound and literature documents well the outlined research activities.

I personally appreciate the topic and the content of the article. Nowadays, spatial justice and socio-spatial inequalities are challenges, which can be met by awareness building and evaluation of the situation. A conceptual framework to access the spatial justice is a first step to meet these challenges.

Proposals for improvement (by reviewer)

Nevertheless, I see some potential to improve the structure of the article.

  • The introduction describes to some extend the procedural method (… we refer to the IAD), which is documented also in the research methodology (… we use the IAD). I would propose to focus the introduction to the description of challenges and requirements of spatial justice and shift implementation issues to chapter 2.
  • For better understanding, the Theoretical Background could be preposed the research methodology.
  • As most of the last chapter is a summary of the article, this should also be made clear by the title "Summary and conclusions”.
  • Additional conclusions have to be added.
  • Change the numbering system for literature, as it does not comply with the specifications of the journal.
  • References 17, 71, 73, 74 and 82 not quoted in text.

Proposals for improvement - Detailed comments

  • Substitute “to construct indicators” by “to elaborate indicators” or “to develop indicators”
  • Chapter 1 / Paragraph 3 / Line 2: Capital letter for “Development” & add “(IAD)”, as this abbreviation is used in line 4.
  • Figure 1 and Figure 2: use plural also for “rules” (Rules-in-use)
  • Chapter 4.1/ P1/L8: Delete “a” in the phrase “consider the project in a different scales” or delete “s” in the word “…scales”
  • Table 2: Delete dots in the column “Indicator” (three times)
  • Table 2: Check wording “meet the needs in the neighbors”
  • Chapter 4.2/P1/L16: Check reference: Peter Marcuse [10] emphasises…
  • Chapter 4.3/P3/L2: Add “s” in the phrase “This highlight_ the …”
  • Table 4: Delete dot in the column “Indicator” (one time)
  • Table 4: Check width of end line
  • Chapter 4.3/P6/L1: Add “s” in the phrase “This framework concern_ …”
  • Chapter 4.3/P6/L4: Delete “s” in the phrase “…the indicators needs ….”

Additional comment

  • The paper was checked against plagiatism. No violations were detected

Round 2

Reviewer 1 Report

No comments